# Aptamer-Functionalized Iron-Based Metal–Organic Frameworks (MOFs) for Synergistic Cascade Cancer Chemotherapy and Chemodynamic Therapy

**DOI:** 10.3390/molecules27134247

**Published:** 2022-06-30

**Authors:** Xuan Wang, Qing Chen, Congxiao Lu

**Affiliations:** 1School of Pharmacy, Qingdao University, Qingdao 266021, China; wangxuan5386@163.com; 2Department of Pharmacy, Zaozhuang Municipal Hospital, Zaozhuang 277102, China; 3College of Life Sciences, Zaozhuang University, Zaozhuang 277160, China; 4Department of Pharmacy, Yantai Yuhuangding Hospital, Yantai 264099, China

**Keywords:** aptamer, MOFs, chemodynamic, chemotherapy, breast cancer

## Abstract

Hypoxia-activated prodrugs (HAPs) with selective toxicity in tumor hypoxic microenvironments are a new strategy for tumor treatment with fewer side effects. Nonetheless, the deficiency of tumor tissue enrichment and tumor hypoxia greatly affect the therapeutic effect of HAPs. Herein, we design an active targeted drug delivery system driven by AS1411 aptamer to improve the tumor tissue enrichment of HAPs. The drug delivery system, called TPZ@Apt-MOF (TA-MOF), uses iron-based MOF as a carrier, surface-modified nucleolin aptamer AS1411, and the internal loaded hypoxia activation prodrug TPZ. Compared with naked MOF, the AS1411-modified MOF showed a better tumor targeting effect both in vitro and in vivo. MOF is driven by GSH to degrade within the tumor, producing Fe^2+^, and releasing the cargo. This process leads to a high consumption of the tumor protective agent GSH. Then, the Fenton reaction mediated by Fe^2+^ not only consumes the intracellular oxygen but also increases the intracellular production of highly toxic superoxide anions. This enhances the toxicity and therapeutic effect of TPZ. This study provides a new therapeutic strategy for cancer treatment.

## 1. Introduction

According to the World Health Organization, breast cancer has the highest incidence rate of cancers among women worldwide, reaching sixteen percent of all female cancers. In the USA, female breast cancer has reached epidemic proportions, the second most common cause of death in this country [1]. Despite the development of multiple therapies including surgery, radiotherapy, and chemotherapy, the treatment of breast cancer, especially advanced breast cancer, remains a clinical challenge. Systemic toxicity and treatment tolerance are the most pressing issues.

Hypoxia-activated prodrugs (HAPS) are a selective toxic agent in a hypoxic environment that have been vigorously studied; they have broad tumor therapeutic prospects [2,3]. Tirapazamine (TPZ), a representative HAP, is up to 300 times more toxic under hypoxia than under aerobic conditions in mouse and human cancer cell lines [4]. As a precursor drug, TPZ can produce instantaneous oxidizing radical intermediates through a one-electron reduction reaction under hypoxia conditions. The dehydration of TPZ oxidizing radical intermediates can produce benzotriazine (BTZ) radicals with cytotoxicity [5]. BTZ radicals can interact with intracellular DNA to cause structural damage and induce cell apoptosis [6]. Although phase I and II studies showed that TPZ had considerable chemotherapeutic potential, phase III studies failed to show additional benefits of TPZ due to the failed accumulation of TPZ in tumors [7,8,9]. Therefore, for TPZ, the development of a targeted drug delivery system can improve its antitumor effect. A nano drug delivery system can achieve targeted drug delivery, reduce a drug’s toxic and side effects, decrease the total dose, and improve the bioavailability of insoluble drugs [10,11,12,13,14]. By modifying the surface of nanocarriers with specific ligands, such as antibodies, the drug or active ingredient can be delivered to tumor tissues, resulting in higher activity selectivity [15,16,17,18]. In addition, inadequate tumor hypoxia is another key cause of TPZ treatment failure [19]. If tumor hypoxia can be selectively increased, the therapeutic effect of TPZ may be enhanced.

Chemodynamic therapy (CDT) is an emerging tumor therapy strategy that converts hydrogen peroxide (H_2_O_2_) into the most toxic reactive oxygen species (ROS) -hydroxyl radical (•OH) at the tumor site through a Fenton/Fenton-like reaction, thus inducing apoptosis and necrosis of tumor cells [20,21,22,23]. Compared with traditional therapies, CDT has a higher tumor specificity and selectivity and lower systemic toxicity. Iron-based materials are efficient and inexpensive Fenton reagents and are widely used in CDT because iron is a microelement in the body. Among them, iron-based metal–organic frameworks (MOFs) have the following advantages: (1) a large pore surface with high drug adsorption capacity; (2) susceptibility to chemical modification, and the pores and surfaces can have specific host–guest interaction; and (3) good biodegradability and biocompatibility [24,25,26,27,28,29]. These advantages indicate that iron-based MOFs have a broad application prospect in the field of tumor comprehensive therapy. Fan et al. reported that MOFs can release loads in response to GSH and simultaneously produce divalent iron ions [30]. Divalent iron ions can catalyze the transformation of the excess H_2_O_2_ produced by cancer cells into highly toxic hydroxyl radicals. Meanwhile, the divalent iron can react with the oxygen in cells to further consume the oxygen in tumors and aggravate the hypoxia state. Therefore, iron-based MOFs are ideal drug delivery carriers for HAPs, which can improve the tumor enrichment of drugs through targeted modification, while releasing iron ions mediates the Fenton reaction and oxygen consumption to enhance the antitumor effect of TPZ.

In this study, an AS1411 aptamer-functionalized iron-based MOF called TPZ@Apt-MOF (TA-MOF) was synthesized for tumor tissue targeted delivery of TPZ. As shown in Figure 1, TA-MOF can actively target breast cancer and be internalized by cancer cells. The tumor microenvironment drives its degradation, which releases iron divalent ions and TPZ. Fenton reactions mediated by the iron divalent ions produce large amounts of hydroxyl radicals and consume oxygen and GSH. This process increases tumor hypoxia and drug sensitivity, thus improving the anticancer efficacy of TPZ. This study provides a new direction for the research into a new generation of tumor therapeutic agents through the construction of targeted cascade chemodynamic synergistic hypoxic chemotherapy nanosystems.

## 2. Materials and Methods

Materials: We bought 2-amino terephthalic acid, FeCl_3_·6H_2_O, and DMF from Shanghai Aladdin (China). TPZ, NSH, and EDC were purchased from Shanghai Macklin (China). AS1411 was purchased from Shanghai Sangon Biotech (China). An ROS Assay Kit and Cell Counting Kit-8 were purchased from Beyotime Biotechnology (China). DMEM medium and Trypsin were obtained from Thermo Fisher Scientific (USA). Fetal Bovine Serum was purchased from Gibco (USA). An Annexin V-FITC/PI Apoptosis Detection Kit and DAPI were purchased from Beijing Solarbio (China).

Animals and Cells: Mouse breast cancer 4T1 cells were purchased from Shanghai Mingjin Biotechnology Co., Ltd., (China). The cells involved in the experiment were cultured in 1640 medium or DMEM medium containing 10% fetal bovine serum. The required conditions are for the incubator to reach 37 °C and contain 5% CO_2_. Balb/c female mice (4–5 weeks) were taken from the experimental animal center of Shandong University. The experiments on the mice were carried out with the permission of the Ethics Committee of Zaozhuang University.

The preparation of MOF: The preparation of MOF was based on previous literature reports [31]. In brief, we dissolved 2.08 mmol FeCl_3_·6H_2_O and 1.03 mmol 2-aminoterephthalic acid in 90 mL DMF, and 3.6 mL acetic acid was added to the mixture. The mixture was dispersed evenly and treated at 110 °C for 10 h. After the reaction, the mixture was centrifuged at 8500 rpm for 15 min to obtain the brick-red solid product NH_2_-MOF (Fe). Then, the obtained solid product was washed with DMF and methanol three times, respectively; the resulting MOF (Fe) was dispersed in DMF for further use.

Construction of Apt-MOF and TA-MOF: First, 1 μmol NSH, 1 μmol EDC, and 0.1 mg AS1411 aptamer were dissolved in 1 mL DMF and stirred until fully mixed and activated at room temperature for 1 h. Then, 10 mg MOF was added, stirred at room temperature for 24 h, centrifuged (8500 rpm, 10 min), and cleaned with DMF three times to obtain AS1411@MOF (denoted as Apt-MOF). Next, the centrifuged supernatant fluid was collected, and the content of free AS1411 aptamer was determined by a NanoDrop spectrophotometer. Then, 1 mg TPZ and 1 mg apt-MOF were added to 2 mL ddH_2_O. At room temperature, we stirred the mixture overnight. Finally, the mixture was centrifuged to obtain TPZ@Apt-MOF (denoted as TA-MOF). The supernatant was used to measure the encapsulation rate. The drug loading capacity of TPZ was determined by UV–vis spectrophotometry, and the final efficiency was about 85%. Refer to the previous literature for the specific methods [32]. 

Characterization: After the preparation, we diluted the MOF with ultrapure water, then directly dropped it into the copper mesh covered with carbon film, and observed it with TEM. MOF is a kind of nanoparticle, and we used Malvern nano ZS90 to test its zeta potentials and particle size. FTIR was used to detect the formation of the amide bond.

Cellular internalization: In order to observe the cellular internalization of MOF, we referred to the previous paper, but in a slightly different way [33]. In brief, cells were washed by PBS 2–3 times for 2–3 min each time. After changing the medium and adding nanoparticles at the same time, the cells were co-incubated at 37 °C for a specified time. The free nanoparticles were then removed, and the cells were immobilized with 4% paraformaldehyde for 10 min before staining the cytoskeleton and nucleus. After dyeing, the free dyes were washed away, and the fluorescence distribution was observed by laser scanning confocal microscope (LSCM).

Cell viability assay: By referring to the previous literature reports, we used the CCK-8 method to detect the viability of tumor cells [33]. In brief, 4T1 cells were seeded into 96-well plates with 4000 cells in each well and incubated overnight in an incubator with 5% CO_2_ at 37 °C. The prepared solutions (TPZ, Apt@MOF, TA-MOF, with different concentration gradients) were added to each well, respectively, and the cells were cultured for 24 h. The old medium was removed; then, the CCK8 working solution was added, and the cells were cultured for another 30 min. Finally, the absorbance was measured at 450 nm with a microplate reader.

Apoptosis detection: The instructions for the Annexin V-FITC/PI apoptosis detection kit provided us with the detailed operation methods. In brief, after drug intervention, the 4T1 cells were washed twice with pre-cooled PBS and collected after centrifugation for 5 min. After the supernatant was discarded, the 4T1 cells were resuspended by adding 500 μL Binding Buffer. Then, 5 μL Annexin V-FITC and PI were added and mixed gently. The samples were incubated for 10–15 min at room temperature under dark conditions and detected by flow cytometry within 1 h.

Detection of intracellular ROS levels: Intracellular reactive oxygen species (ROS) were detected by using 2, 7-dichlorofluorescence-based yellow diacetate (DCFH-DA). In short, 4T1 cells were inoculated into 12-well plates with about 6000 cells per well. After overnight culture, the drugs of the different experimental groups (TPZ, Apt@MOF, TA-MOF) were co-incubated with 4T1 cells for 24 h. Then, DCFH-DA was added and incubated for 30 min under standard conditions. Finally, after the staining was completed, we could directly observe and take photos under a laser confocal microscope.

Animal Imaging Experiments: The IVIS Spectrum was used to monitor the distribution of the ICG fluorescence signal in mice to track the metabolism of TA-MOF in vivo [34]. First, 0.15 mg TA-MOF was injected into each mouse through the tail vein. Then, the relationship between the fluorescence distribution and time was observed through a small animal fluorescence imaging system. 

Therapeutic effect experiment: The tumor-bearing mice were constructed subcutaneously, and the tumor volume was collected on day 7 when the tumor reached about 80 mm^3^ in size. All dosing was performed through a tail vein every two days for three consecutive dosing sessions. The tumor volume was counted up to 15 days. The survival rate of the mice was measured at the same time. Specifically, the survival and tumor size of mice were observed, and the death of a mouse or a tumor diameter exceeding 2 cm was considered as treatment failure.

H&E Staining: H&E staining was conducted, referring to the previous literature reports [35]. In order to evaluate the safety of the nanomedicine, the major organs of the mice in each group were sampled after administration. After that, we needed a sufficient amount of fixative 4% formaldehyde to fix the extracted mouse organs and tissues overnight. The tissue section specimens were prepared by the paraffin-embedded sectioning method before H&E staining. Finally, Motic Easy Scan was used to observe and analyze all sections.

Statistical Analysis: Statistical analysis was performed with Student’s paired *t*-test (SPSS Inc., Chicago, IL, USA), and the differences were considered to be statistically significant at *p* < 0.05. Kaplan–Meier survival analysis was used to analyze the survival rate of the tumor-bearing mice.

## 3. Results and Discussion

### 3.1. Construction and Characterization of TA-MOF

The EPR effect, namely the enhanced permeability and retention effect of solid tumors, allows nanoparticles to passively target tumor tissues, which is the premise of active targeting. In view of the particle size requirement of the EPR effect (30–200 nm), we synthesized MOF with a particle size of about 100 nm. Transmission electron microscopy (TEM) (Figure 1A) showed that the nanocarrier had uniform morphology and a clear crystal structure, indicating that the synthesis was successful. To obtain a more complete picture of the particle size distribution, the dynamic light scattering (DLS) technique was used to measure the overall particle size distribution of the synthesized nanoparticles, and the experimental results (Figure 1B) were consistent with those obtained by electron microscopy. Then, we performed targeted modification of the MOF by coupling the AS1411 on the surface. To verify the linkage, we used Fourier transform infrared spectroscopy (FTIR) to determine the formation of amide bonds. As shown in Figure 1C, the modified carrier appeared as a characteristic peak at the position, indicating that the link was successful. Meanwhile, the surface potential was measured to further confirm the modification results. As shown in Figure 1D, the surface potential of the nanocarrier was positive originally but became negative after modification by AS1411. According to the published studies, it is speculated that that the combination of the negatively charged nucleic acid with the surface of the nanocarrier leads to the change in the surface potential. These results confirm that the AS1411 was linked to the surface of the MOF.

### 3.2. Tumor Targeting Effect of TA-MOF

To verify the tumor targeting effect of TA-MOF, LSCM imaging was used to compare the effect of MOF with or without aptamer modification on targeting 4T1 cells. As shown in Figure 2A,B, the MOF without AS1411 modification was internalized by tumor cells, but under the same conditions, its internalization efficiency was much lower than that of the TA-MOF group. This experimental result indicated that AS1411 modification increased the targeting effect of tumors. This result is consistent with previous reports [36,37,38] and preliminarily demonstrates the targeting of the TA-MOF. To verify the tumor targeting effect of the TA-MOF in vivo, a mouse model of a subcutaneous tumor was constructed for NIR fluorescence imaging. As shown in Figure 2C,D, the nanoparticles enriched in the tumor tissue and maintained for a long time. After intravenous injection, the fluorescence of the tumor tissue appeared at 12 h and reached its maximum intensity at 24 h. The above experiments proved that the nanoparticles had good tumor targeting activity at the cellular and animal level.

### 3.3. Antitumor Effect of TA-MOF In Vitro

To verify the antitumor effect of TA-MOF, 4T1 mammary carcinoma were used to measure changes in cell viability after treatment with TA-MOF. The results are shown in Figure 3A; the TA-MOF group had a significant inhibitory effect compared with the free TPZ group and the MOF group. At the same time, cell apoptosis was detected, as shown in Figure 3B, which also significantly increased more than the free TPZ group and the MOF group. We hypothesize that the iron divalent produced during the MOF metabolism enhanced the chemotherapy toxicity of the TPZ. For this reason, we detected the production of ROS in cells (Figure 3C). The TA-MOF group induced a significant increase in ROS, and the cells exhibited stronger fluorescence brightness than the free TPZ and MOF groups. In conclusion, TA-MOF can promote ROS production through the synergy of the Fenton response and hypoxic prodrugs, thus inducing tumor cell apoptosis.

### 3.4. Antitumor Effect of TA-MOF In Vivo

We tested the in vivo antitumor effect of TA-MOF by constructing a 4T1 tumor-bearing mouse model, and the tumor volume and survival rate were analyzed. The experimental results are shown in Figure 4A; the tumor volume was effectively inhibited. The survival rate of the mice was significantly increased (Figure 4B). The average survival time of the mice in the TA-MOF group was 48 days, while that of the mice in control group was 28 days. In conclusion, TA-MOF has a good in vivo antitumor effect, can significantly inhibit the tumor volume growth, and improve the survival rate.

### 3.5. Tissue Toxicity of TA-MOF

To verify the safety of the material, we performed an autopsy on mice 3 days after administration and obtained the major organs for H&E staining. The experimental results showed (Figure 5) that there was no obvious damage to the major organs. The experimental results showed that the nanoparticles had no obvious tissue toxicity, which was consistent with the previous study of MOF as a carrier. 

## 4. Conclusions

TA-MOF achieved tumor tissue enrichment through the targeting effect mediated by AS1411 aptamer. Subsequently, the nanocarrier was degraded by the tumor microenvironment, releasing iron divalent and TPZ. The Fenton reaction mediated by the iron divalent played a role in chemodynamic chemotherapy, simultaneously consumed intracellular oxygen and reduced glutathione, exacerbated tumor hypoxia and drug sensitivity, and thus realized tumor cell sensitization to TPZ. This targeted delivery and tumor microenvironment response ensures the accurate delivery and controlled release of the TPZ. At the same time, the Fenton reaction-mediated oxygen consumption and production of a highly toxic hydroxyl radical can sensitize the chemotherapy effect of TPZ. This study may provide a promising targeted collaborative therapy strategy for future cancer treatment.

## Data Availability

The data in the paper can be obtained from the authors.

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
