# Peer review of "Aptamer-Functionalized Iron-Based Metal–Organic Frameworks (MOFs) for Synergistic Cascade Cancer Chemotherapy and Chemodynamic Therapy"

_molecules, 2022, doi:10.3390/molecules27134247_

Round 1

Reviewer 1 Report

This is an interesting work with potential practical applications. I recommend publication.

Just one minor comment before publication:

Many abbreviations are used in the manuscript which makes difficult to read the ms. A  list of abbreviations must be inserted.

Reviewer 2 Report

The paper of Congxiao Lu and co-authors is a fundamental work on synthesis iron-based MOF (previously described), modified it with anticancer agent and testing the latter on antitumor effect (mise and cell lines). MOF-based drug delivery systems are in frontier and I recommend the paper to publish in Molecules. 

What are the symbols on figure 2 B (### ****)

What are the symbols on figure 3 A (# ** ***)

I didn’t find legend or information in the text

I didn’t check all literature sources but look at the ref 13. “The preparation of MOF was based on previous literature reports [13]” , Line 102. But ref 13 is not about MOF but “graphene nanosystem” which does not contain Iron atoms. Is it a mistake…? 

Reviewer 3 Report

This paper describes the synthesis and characterization of an aptamer functionalists Fe MOF and it’s targeted delivery property of the prodrug TBZ. This approach is especially efficient for the breast cancer therapy. Biological studies were performed proving that the functionalised MOF is more efficient than the pristine one. This is an interesting piece of work that can be published in Molecules after the authors have considered the below comments. 1. Thermal stability studies can be performed to the pristine MOF and the functionalised one to confirm that the stability and porosity of the MOF are not affected after functionalisation. 2. Have the conditions for the optimum drug uptake been investigated? For example how the ratio of drug:functionalised MOF affect the uptake? 3. How the drug is stabilized in the MOF? Is it encapsulated into the MOF pores or on to its surface?
